# Bayesian Nonparametric Survival Analysis via Deep Dirichlet Process

## Abstract

The analysis of time-to-event data has received increasing attention in many application fields. The key challenge is that the data are mostly incomplete, with the right censoring mechanism being the most popular form. While Cox's proportional hazards assumption has shown adaptivity to traditional time-to-event datasets, challenges are observed when generalizing this assumption to modern survival analysis — the proportional hazards assumption is often violated when covariates are high-dimensional. Moreover, traditional parametric assumptions on the survival distribution mostly belong to the exponential family and thus the assumption is strong and their exponential decay rate leads to poor long-tail approximations. To overcome these challenges, we propose a novel deep learning framework for survival analysis, named **DDPSurv**, which adopts a deeply parameterized Dirichlet process (DP) mixture model on survival distribution. Different from previous deep parametric approaches which rely on strong statistical assumptions, our framework can model the survival distribution with greater flexibility by adopting a DP mixture model. With the DP mixture model, we can improve the flexibility in modelling the survival distributions and achieve better tail behaviour by including the heavy-tail distributions in the mixture. We theoretically show that the proposed model can approximate the true survival distribution at a tight concentration rate. Empirical evaluations on standard survival benchmarks validate the satisfactory performance of the proposed method. The extensive experiments on large-scale clinical datasets — MIMIC-III and MIMIC-IV — highlight the scalability and clinical significance of our method. Codes are anonymously available at `https://anonymous.4open.science/r/DeepSurv-net-2215`

## 1 Introduction

One of the key challenges that differentiate time-to-event data from other types of data is the incomplete data issue and the censoring mechanism, including right-censoring (Nagpal et al., 2021b; Cox, 1972), and interval censoring (Zhang and Yin, 2022). Survival analysis, a branch of time-to-event modeling, deals with right-censored data in most cases and has found applications in various domains, such as clinical trials (Cox, 1972; Zhang and Yin, 2022) and actuarial science (Logubayom and Yeboah, 2023). Despite the success of traditional survival analysis statistical models (Gu et al., 2023; Li et al., 2018), they are incapable of capturing the significant features when scaling to modern high-dimensional datasets. Therefore, it is emerging to introduce deep learning approaches to time-to-event modeling to deal with high-dimensional data (Katzman et al., 2018; Faraggi and Simon, 1995; Xiang et al., 2000), as deep neural networks allow for a more flexible nonlinear relation between covariates and hazard ratio.

Although deep neural networks have significantly improved survival models' prediction accuracy and flexibility, they now face the new challenge of being over-parameterized, leading to potential over-fitting issues. Recently, probabilistic models have been incorporated into the learning objective to regularize the learning process, where the deep Cox model (Nagpal et al., 2021b) is a well-known method that integrates the Cox proportional hazards (CPH) model and deep learning. However, these existing models rely on strong parametric assumptions (e.g., exponential family) and lack flexibility in modeling the survival distribution. Especially when the observations in tails are limited and the exponential tail decay would mostly lead to bias at long tail. Moreover, these works are heavily based on the CPH assumption with a negative partial likelihood to address the challenge of missing

data, making it difficult to handle the cross-hazard scenario that is very common in real data analysis (Mantel and Stablein, 1988).

In this paper, we propose DDPSurv, a deep Dirichlet process model with heavy tail mixtures, to estimate the survival distribution more accurately and efficiently. Relying on the richer-gets-richer property of the Dirichlet process, our framework can automatically find the number of mixtures and be able to select the most prominent distribution from the infinite number of mixtures, providing the best approximation of the survival distribution.

Our contributions can be summarized as follows: (1) We propose a new neural survival analysis framework (DDPSurv) based on a deep Dirichlet process, which can tackle survival prediction at high dimensions. (2) By mixing heavy tail distributions, we validate that the DP model can better approximate the survival distribution at tails and mitigate the long-tail bias. DDPSurv can also tackle the competing risk scenarios since each distribution in the infinite mixture can model the hazard rate of a particular risk. (3) DDPSurv adopts stochastic variational inference to avoid the high computational cost in parameter estimation via existing sampling-based methods (Zhang and Yin, 2023; Müller et al., 2015) (e.g., Gibbs sampling), and enables scalability to large-scale datasets. (4) Theoretical analysis under the Sieve space framework provides asymptotic bounds to the posterior concentration rate of the parameters. (5) Extensive experiments on generic survival predictions and two large-scale clinical datasets validate the satisfactory performance of our proposed method.

## 2 RELATED WORKS

**Survival Analysis.** Time-to-event modeling, particularly in the presence of censoring data, has been an important topic in statistical prediction across various domains such as economics (Bosco Sabuhoro et al., 2006; Jones et al., 2002), actuarial science (Czado and Rudolph, 2002), and medical treatment (Zhu et al., 2016; Kim et al., 2019). Survival analysis, a major subfield of time-to-event modeling, has been extensively studied. Two major traditional parametric or semi-parametric survival models that have played a prominent role in survival analysis are the Cox proportional hazard model (CPH) (Cox, 1972) and the accelerated failure time model (AFT) (Wei, 1992). A substantial body of literature (Kraisangka and Druzdzel, 2016; 2018; Rosen and Tanner, 1999) has focused on improving these models to achieve higher prediction performance.

In recent years, deep neural networks and stochastic variational inference methods have been applied to enhance traditional parametric or semi-parametric survival analysis (Nagpal et al., 2021b; Katzman et al., 2018; Alaa and van der Schaar, 2017; Zhong et al., 2021) to further improve estimation performance. While the deep neural networks framework increases the flexibility of the model and improves its capacity to handle high-dimensional data problems, stochastic variational inference allows the model to backpropagate gradients and thus save computational costs. DeepHit (Lee et al., 2018) and deep survival machines (DSM) (Nagpal et al., 2021a) have successfully learned fully parametric models while employing stochastic variational inference. However, these models still have limitations due to their fixed parameter settings or model assumptions including the number of mixture components in DSM and discrete-time cases and single-death causes in DeepHit. DSM, in particular, is known for its ability to handle competing risks by learning shared representations. Noting the limitations of previous parametric deep learning models for survival analysis, neural frailty machine (NFM) (Wu et al., 2023) manages to build a fully parameterized deep learning model based on frailty-based statistic models and provides robust statistical theoretical analysis for its convergence of prediction bias. NFM does not use a mixture model structure and still lacks flexibility when approximating survival functions.

**Non-Parametric Analysis.** Non-parametric models have played a crucial role in statistical analysis, offering flexibility and wide applicability (Satagopan et al., 2004; Peterson, 2009; Steinwart and Christmann, 2008). In the field of survival analysis, traditional non-parametric methods are mainly frequentist methods, including the Kaplan–Meier (KM) estimator (Kaplan and Meier, 1958) and the Nelson–Aalen estimator (Nelson, 1969). The KM estimator approximates the survival function by adjusting for the observed event times in its immediate neighborhood. However, frequentist methods ignore the prior knowledge and have a limited function search space compared with Bayesian methods. While Bayesian methods alleviate the limitations of parametric modelling and allow for a larger search space of functions. Among these Bayesian non-parametric methods, the Dirichlet process

combined with the Gibbs sampling method has recently been used to solve the survival problems for its outstanding performance in clustering (Zhang and Yin, 2023) due to the richer-gets-richer property. However, this method still relies on a traditional statistical approach using a Dirichlet process of small-size parameters, without incorporating deep neural networks, which may limit its ability to handle complex features and high-dimensional data. Additionally, the Gibbs sampler may perform worse than the variational inference method in terms of computing efficiency. Therefore, stochastic variational inference needs to be adopted to incorporate non-parametric model into modern deep learning settings.

**Long-Tail Bias Correction.** The problem of long-tail bias has been challenging for many real datasets in insurance, healthcare, and survival analysis. (Fackrell, 2009; Hakim et al., 2021) Since the observations at tails are rather limited in most of the scenarios, it is difficult to approximate the distributions at tails. Previous works have employed a large number of parametric distributions belonging to an exponential family (Gardiner et al., 2014; Hakim et al., 2021) to handle the survival distribution when the data are heavy-tailed. However, common primitive distributions used in survival analysis (e.g., Weibull, log-normal) have poor tail behaviours due to their exponential tail decay (Landsman and Tsanakas, 2012).

Recently, two trends tackling the drawbacks of exponential family distributions have been proposed. One is to re-balance the dataset before the model learns representations by oversampling the tail data, augmenting tail data, or under-sampling the head data (Buda et al., 2018; Beery et al., 2020).

Another one is to solve the long-tail bias by re-weighting the loss, setting the loss to be non-uniform, to facilitate learning the tail data (Cui et al., 2019; Samuel and Chechik, 2021). These approaches mainly focus on adjusting the dataset and the loss. They only provide a universal correction on the distribution and are still dominated by the exponential tail decay (e.g., Nagpal et al. (2021a)). Therefore, a more flexible correction that can adaptively determine the density adjustment is needed for a more accurate approximation at tails.

Figure 1: Our proposed survival model in plate notations.

## 3 METHODOLOGY

### 3.1 PRELIMINARIES

**Problem: Time-to-event Modelling.** We assume a right-censoring model for simplicity. Let $\mathcal{D} = \{(\boldsymbol{x}_i, t_i, \delta_i)\}_{i=1}^n$ denote the dataset as a set of tuples, where $\boldsymbol{x}_i \in \mathbb{R}^d$ is the features associated with individual $i$, $t_i$ is the time at which an event of interest occurs, or the time of censorship, and $\delta_i$ is the indicator that specifies whether $t_i$ is the event time or censoring time. We denote the uncensored subset of $\mathcal{D}$ as $\mathcal{D}_U$ and the censored subset as $\mathcal{D}_C$.

**Definition 3.1** (Dirichlet Process). Denoted as $\mathrm{DP}(\alpha, G)$, the Dirichlet process is a random probability measure on the sample space $\mathcal{X}$, such that for any measurable finite partition of $S$, denoted as $\{B_i\}_{i=1}^K$,

$$(X(B_1), X(B_2), \ldots, X(B_K)) \sim \mathrm{Dir}(\alpha G(B_1), \alpha G(B_2) \ldots, \alpha G(B_K))$$

**Definition 3.2** (Dirichlet Process Mixture (DPM)). Let $\mathrm{DP}(\alpha G_0)$ denote a Dirichlet process with parameter $\alpha G_0$ where $\alpha$ is a precision parameter and $G_0$ is a base probability distribution. The DPM model is defined as

$$G \sim \mathrm{DP}(\alpha G_0),$$
$$\theta_1, \ldots, \theta_T \sim G,$$
$$x_k | \boldsymbol{\theta}_k \sim f_{\boldsymbol{\theta}_k},$$

where $T$ is the truncated number of mixtures.

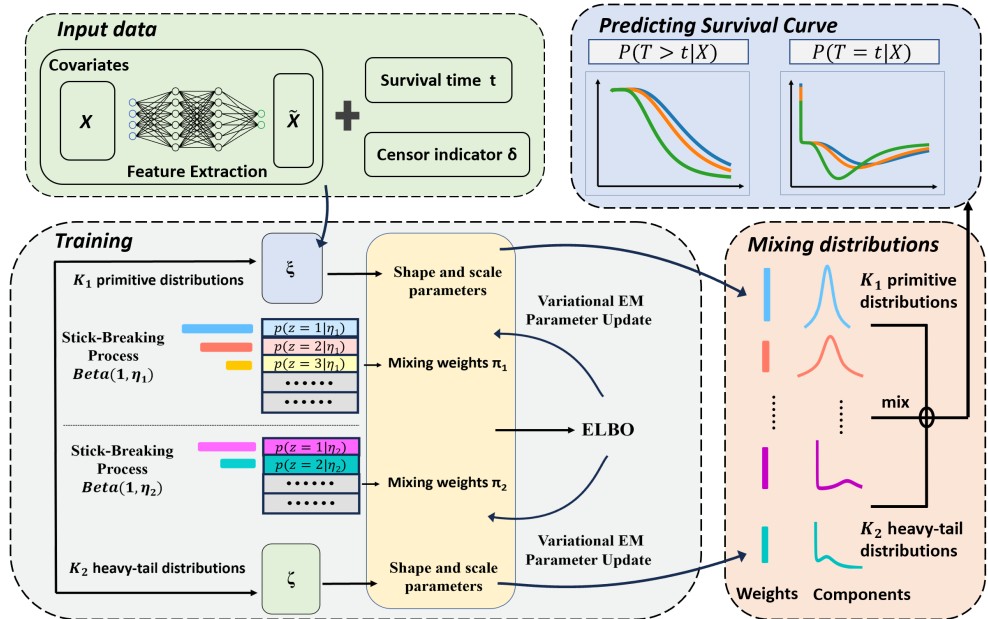

Figure 2: The workflow of our proposed survival analysis framework. We first encoder the scale and shape parameters of the primitive distributions. We then adopt the stick-breaking formulation of the Dirichlet process to determine the mixture weights of each mixture component.

**Definition 3.3** (Heavy-tail Distribution). We consider a distribution $F(x)$ as heavy-tail if its tail convergence rate is slower than an exponential decay, as given by

$$\int e^{tx} dF(x) = \infty, \text{ for all } t > 0.$$

### 3.2 DEEP DIRICHLET PROCESS MIXTURE MODELS

We represent the survival function with a mixture of primitive distributions (e.g., Log-Normal, Weibull), where detailed description of common primitive distributions is provided in the appendix. We specify the Dirichlet process mixture model with the well-known stick-breaking formulation,

$$\alpha_k \sim \text{Beta}(1, \eta_1), \qquad \pi_k = \alpha_l \prod_{k=1}^{K_1-1} (1 - \alpha_k), \qquad (1)$$

where $\pi_k$ is the probability assigned to each cluster. We further assume for each individual we either observe the actual failure time or censoring time but not both. Then we have $z|\pi \sim \text{Cat}(\cdot|\pi)$ the cluster assignment sampled from the multinomial distribution based on the cluster probability $\pi$, and $\eta_1$ is the concentration hyperparameter. We truncate the number of primitive distributions at $K = K_1$.

**Primitive Distributions.** Let $\boldsymbol{\xi} = \{\boldsymbol{\xi}_k\}_{k=1}^{K_1}$ represent the set of all shape and scale parameters of the primitive distribution. We adopt the log-normal distribution as an illustrative example for the primitive distribution where $\boldsymbol{\xi}_k = (\vartheta_k, \varsigma_k)$. Then density $f(t|\boldsymbol{\xi})$ and the survival function $S(t|\boldsymbol{\xi})$ are given by

$$f(t|\boldsymbol{\xi}_k) = \frac{1}{t\varsigma_k\sqrt{2\pi}} e^{-\frac{(\log t - \vartheta_k)^2}{2\varsigma_k^2}}, \qquad S(t|\boldsymbol{\xi}_k) = 1 - \frac{1}{2}\text{erfc}\left(-\frac{\log t - \vartheta_k}{\sqrt{2}\varsigma_k}\right).$$

Further discussions on the primitive distributions are presented in the Appendix.

**Mitigating Long-tail Bias.** As the existing distributions based on the exponential family suffer from exponential tail decay and poorly model the tail behaviour of the true survival distribution, we

---

**Algorithm 1** Our proposed DDPSurv framework.

**Input:**
Data $\mathcal{D} = \{(x_i, t_i, \delta_i)\}_{i=1}^n$
$K_1$, the maximum number of primitive distributions;
$K_2$, the maximum number of heavy-tail distributions;
Parameter sets of primitive distributions $\boldsymbol{\xi}$ and heavy-tail distributions $\boldsymbol{\zeta}$.
Hyperparameters $\{\eta_1, \eta_2\}$,
Parameter sets of networks $\boldsymbol{\Psi} = \{\psi_k\}_{k=1}^{K_1}$ and $\boldsymbol{\Upsilon} = \{\upsilon_k\}_{k=1}^{K_2}$ to encode the shape and scale parameters of the primitive distributions and heavy-tail distributions, respectively.
**Output:** Trained $\boldsymbol{\Psi}$ and $\boldsymbol{\Upsilon}$
1: Sample $\pi_k$ with Eq. (1)
2: **for** each training epoch **do**
3:     **for** $g \in \{1, \ldots, K_1\}$ **do**                                   ▷ Primitive Distributions
4:         Sample mixture weights by Eq. (1) with concentration rate $\eta_1$
5:         Encode parameters $\boldsymbol{\xi}$ with $\boldsymbol{\Psi}$
6:         Compute log-likelihood
7:     **end for**
8:     **for** $g \in \{1, \ldots, K_2\}$ **do**
9:         Sample mixture weights by Eq. (3) with concentration rate $\eta_2$
10:        Encode parameters $\boldsymbol{\zeta}$ with $\boldsymbol{\Upsilon}$
11:        Compute log-likelihood with Eq. (2)
12:     **end for**
13:    Compute ELBO with Eq. (8)
14:    Backpropagate ELBO to $\boldsymbol{\Psi}$ and $\boldsymbol{\Upsilon}$
15: **end for**
16: **return** Trained $\boldsymbol{\Psi}$ and $\boldsymbol{\Upsilon}$.

---

further include $K_2$ heavy-tail distributions into the mixture to improve the tail behaviour (Dey and Yan, 2016; Ibragimov et al., 2015). Without loss of generality, we adopt the log-Cauchy distribution with density function

$$f(x; \mu, \sigma) = \frac{1}{x\pi\sigma \left[1 + \left(\dfrac{\log x - \mu}{\sigma}\right)^2\right]}, \tag{2}$$

which has a logarithmically decaying tail. In addition to the mixture of primitive distributions, we mix infinite numbers of heavy tail distributions as specified by Eq. (2). We then apply another stick-breaking process to compute the mixture weights $\{\lambda_k\}_{k=1}^{K_2}$,

$$\beta_k \sim \text{Beta}(1, \eta_2), \qquad\qquad \lambda_k = \beta_l \prod_{k=1}^{K_2-1} (1 - \beta_k). \tag{3}$$

Let $\boldsymbol{\zeta} = \{\boldsymbol{\zeta}_k\}_{k=1}^{K_2}$ represent the sets of all shape and scale parameters of the heavy-tail distributions, where $\boldsymbol{\zeta}_k$ is the set of parameters of the $k$-th mixture. We further denote $\phi = \{\boldsymbol{\xi}, \boldsymbol{\zeta}\}$.

### 3.3 STOCHASTIC VARIATIONAL INFERENCE

Without loss of generality, we focus on the right-censoring scheme, and we specify the censoring and uncensoring loss functions as follows.

**Uncensoring Loss.** Given the DP mixture model, we have the following loss for uncensored data,

$$\log \mathbb{P}(D_U | \Theta) = \log \left( \prod_{i=1}^{|D_U|} P(T = t_i | \boldsymbol{x}_i, \Theta) \right) \tag{4}$$

$$= \sum_{i=1}^{|D_U|} \log \left( \sum_{k=1}^{K_1+K_2} P(T = t_i | \boldsymbol{x}_i, \boldsymbol{\xi}_k) P(Z = k | \boldsymbol{x}_i, \beta) \right) \tag{5}$$

**Censoring Loss.** For right-censored data, the ELBO is defined by the likelihood to the survival function

$$\log P(D_C|\Theta) = \log\left(\prod_{i=1}^{|D_C|} P(T > t_i|\boldsymbol{x}_i,\Theta)\right) \tag{6}$$

$$= \sum_{i=1}^{|D_C|} \log\left(\sum_{k=1}^{K_1+K_2} P(T > t_i|\boldsymbol{x}_i,\boldsymbol{\zeta}_k)P(Z=k|\boldsymbol{x}_i,\beta)\right). \tag{7}$$

By adopting a mean-field approximation, the variational family of the parameters is given by

$$q(\boldsymbol{\alpha},\boldsymbol{\beta},\boldsymbol{\xi},\boldsymbol{\zeta},t) = \prod_{k=1}^{K_1-1} q(\alpha_k) \prod_{k=1}^{K_2-1} q(\beta_k) \prod_{k=1}^{K_1} q(\boldsymbol{\xi}_k) \prod_{k=1}^{K_2} q(\boldsymbol{\zeta}_k) \prod_{i=1}^{n} q(t_i|\boldsymbol{\alpha},\boldsymbol{\beta},\boldsymbol{\xi},\boldsymbol{\zeta}).$$

The surrogate loss of backpropagation can be obtained by negating the evidence lower bound, which is computed by the log of posterior mixture weights

$$\mathcal{L}(\boldsymbol{\Psi}) = \mathrm{KL}(q(\boldsymbol{\xi})\|p(\boldsymbol{\xi})) + \mathrm{KL}(q(\boldsymbol{\zeta})\|p(\boldsymbol{\zeta})) + \mathrm{KL}(q(\boldsymbol{\alpha})\|p(\boldsymbol{\alpha}))$$
$$+ \mathrm{KL}(q(\boldsymbol{\beta})\|p(\boldsymbol{\beta})) + \sum_i \mathrm{KL}(q(t_i|\boldsymbol{\alpha},\boldsymbol{\beta},\boldsymbol{\xi},\boldsymbol{\zeta})\|p(t_i|\boldsymbol{\alpha},\boldsymbol{\beta},\boldsymbol{\xi},\boldsymbol{\zeta}))), \tag{8}$$

where $\mathrm{KL}(q(t_i|\boldsymbol{\alpha},\boldsymbol{\beta},\boldsymbol{\xi},\boldsymbol{\zeta})\|p(t_i|\boldsymbol{\alpha},\boldsymbol{\beta},\boldsymbol{\xi},\boldsymbol{\zeta}))$ is the censored likelihood which is specified by Equations (6) and (4), then the problem is reduced to learning the likelihood of inputs to the assumed DP model. The detailed parameter update procedure of the variational posterior and the closed-form KL divergences of the primitive distributions can be found in the supplementary materials.

## 4 THEORETICAL ANALYSIS

**Technical Setups.** For technical simplicity, we assume that the hazard rate after mixture is predicted by the neural network, i.e.,

$$h(t|\boldsymbol{x}) = \nu(t,\boldsymbol{x}).$$

Here $\nu$ is an unspecified non-negative function. The survival function can then be represented by $S(t|X) = e^{-\int_0^t e^{\nu(s,\boldsymbol{x})}ds}$. Then the censored log-likelihood can be re-written as

$$l(T,\delta,\boldsymbol{x};\nu) = \delta \log \int_0^T e^{\nu(s,\boldsymbol{x})}ds + \delta\nu(t,\boldsymbol{x}) + \int_0^T e^{\nu(s,\boldsymbol{x})}ds.$$

We demonstrate the theoretical boundedness of the proposed DDPSurv model using the Sieve space (Wellner et al., 2013; Wu et al., 2023), which provides the rates of convergence in the sense of parametric regression. As in previous works (Wellner et al., 2013; Wu et al., 2023), we choose the Hölder ball to represent the function space,

$$W_M^\beta(\mathcal{X}) = \left\{ F : \max_{\alpha:|\alpha|\leq\beta} \operatorname*{ess\,sup}_{x\in\mathcal{X}} |D^\alpha(f(x))| \leq M \right\},$$

where the domain $\mathcal{X}$ is assumed to be a subset of $d$-dimensional euclidean space, $\alpha = (\alpha_1,\ldots,\alpha_d)$ is a $d$-dimensional tuple of nonnegative integers satisfying $|\alpha| = \alpha_1 + \cdots + \alpha_d$ and $D^\alpha f = \frac{\partial^{|\alpha|} f}{\partial x_1^{\alpha_1} \cdots x_d^{\alpha_d}}$ is the weak derivative of $f$. We assume $M$ is a reasonably large constant.

We make the following assumption for the true parameters:

**Condition 1.** *(True Parameter (Wu et al., 2023)) The Euclidean parameter $\theta_0 \in \Theta \subset \mathbb{R}$, and the two function parameters $m_0 \in W_M^\beta(\mathcal{X})([-1,1]^d)$, $h_0 \in W_M^\beta(\mathcal{X})([0,\tau])$, and $\tau > 0$ is the ending time in the theoretical studies in survival analysis.*

**Condition 2.** *(Sieve space) The Sieve space $\mathcal{V}_n$ is constructed as a set of MLPs satisfying $\hat{\nu} \in \mathcal{W}_{M_v}^{\beta}([0,\tau])$, with depth of order $\mathcal{O}(\log n)$ and total number of parameters $\mathcal{O}\left(n^{\frac{d+1}{\beta+d+1}} \log n\right)$. Here, $M_v$ is a sufficiently large constant such that every function in $\mathcal{W}_{M_v}^{\beta}([0,\tau])$ can be accurately approximated by functions inside $\mathcal{V}_n$.*

Let $\nu_0$ be the true parameter and $\hat{\nu}$ be the corresponding estimate. We define $\mathbb{P}_{\hat{\nu}_n, \boldsymbol{x}}$ to be the estimated conditional distribution given $\boldsymbol{x}$ and $\mathbb{P}_{\nu_0, \boldsymbol{x}}$ to be the true conditional distribution. We further define a metric to measure the convergence of the parameter estimate,

$$d\left(\hat{\nu}_n, \nu_0\right) = \sqrt{\mathbb{E}_{x \sim \mathbb{P}_X}\left[H^2\left(\mathbb{P}_{\hat{\nu}_n, \boldsymbol{x}} \| \mathbb{P}_{\nu_0, \boldsymbol{x}}\right)\right]}, \tag{9}$$

where $H^2\left(\mathbb{P}_{\hat{\nu}_n, \boldsymbol{x}} \| \mathbb{P}_{\nu_0, \boldsymbol{x}}\right) = \int(\sqrt{d\mathbb{P}} - \sqrt{d\mathbb{Q}})^2$ is the squared Hellinger distance between the probability distributions $\mathbb{P}$ and $\mathbb{Q}$. We use $\tilde{\mathcal{O}}$ to hide the poly-logarithmic factors in the big-O notation.

Based on the above regularity conditions of the Sieve space, we can state the following theorem on the rate of convergence.

**Theorem 1.** *(Rate of convergence) Under conditions 1 and 2, we have that $d\left(\hat{\nu}_n, \nu_0\right) = \tilde{\mathcal{O}}\left(\frac{\beta}{2\beta+2d+2}\right)$.*

## 5 EXPERIMENTS

### 5.1 DATASETS AND EVALUATION METRICS

We validate our method on two common datasets for survival prediction — SUPPORT and SYN-THETIC. We additionally include two large-scale benchmarks on clinical data — the MIMIC-III dataset which contains ICU visits of 46,520 patients in 11 years, and MIMIC-IV which contains 331,794 discharge summaries from 145,915 patients admitted to the hospital and emergency department at the Beth Israel Deaconess Medical Center in Boston, MA, USA. Table 1 presents the details of the datasets used for empirical evaluations.

We use two standard metrics in survival analysis for evaluating model performance. One is the concordance index (C-index),

$$\text{C-index} = \frac{\Sigma_{i,j}\mathbb{I}_{T_j < T_i}\mathbb{I}_{r_j > r_i}\delta_j}{\Sigma_{i,j}\mathbb{I}_{T_j < T_i}\delta_j},$$

where $r_i$ is the risk score of the $i$-th unit. Larger C-index value indicates good performance. The other metric is Brier score (BS), $BS = \frac{1}{N}\sum_{t=1}^{N}(f_t - o_t)^2$ where $f_t$ is the predicted probability of the event, $o_t$ is the actual outcome of the event at instance $t$ and $N$ is the number of forecasting instances. Smaller BS value indicates good performance. Detailed descriptions of the metrics are provided in the supplementary materials.

Table 1: Summary of Datasets

| **Dataset** | Type | No. Obs. | Feature Dim. | No. Events | No. Censoring |
|---|---|---|---|---|---|
| **SUPPORT** | Single Risk | 9,105 | 38 | 6,201(68.11%) | 2,904(31.89%) |
| **SYNTHETIC** | Multiple Risk | 5,000 | 9 | 4,003(80.06%) | 997(19.94%) |
| **MICMICIII** | Single Risk | 17,814 | 34 | 2,235(12.55%) | 14598(87.45%) |
| **MICMICIV** | Single Risk | 22,913 | 30 | 2,703(11.80%) | 20210(88.20%) |

### 5.2 COMPARED METHODS

We compare our proposed framework to seven competitors — **(1) Cox Proportional Hazards (CPH)** (Cox, 1972): This is the standard semi-parametric model, making the assumption of constant baseline hazard. The features interact with the learnt set of weights in a log-linear fashion in order to determine the hazard for a held-out individual. **(2) DeepCox**: Proposed by (Katzman et al., 2018), DeepSurv

involves learning a non-linear function that describes the relative hazard of a test instance. It makes the familiar assumption of constant baseline hazard, as does CPH. **(3) DeepHit (DH)** (Lee et al., 2018): This approach involves learning the joint distribution of all event times by jointly modelling all competing risks and discretizing the output space of event times. **(4) Deep Cox Mixture (DCM) (Nagpal et al., 2021b):** This model replaces the parameters in CPH with deep neural networks and adopts a mixture model structure **(5) Deep Survival Machines (DSM)** (Nagpal et al., 2021a): This model is mixture of parametric models from lognormal or weilbull distributions. It does not rely on the strong assumption of cox proportional hazard ratio. **(6) Sumo-Net** (Rindt et al., 2022): This model proposes a simple novel survival regression method using a monotonic restriction on the time-dependent weights to optimize right-censored log-likelihood **(7) NFM (Wu et al., 2023):** This model propose a fully parameterized deep learning model based on the frailty model.

## 5.3 Quantitative Results

We mainly evaluate our model and other baselines at two different time horizons, 25% quantile and 50% quantile. The results are presented in Table 2. We find that our method can overall outperform the baseline methods by a satisfactory margin in most of the settings when C-index and brier score are selected as evaluation metric. Specifically,our DeepSurv ranks first under most of the cases (13/16) and rank second or third for the few other cases. In particular, our model has larger improvements on MIMIC-III and MIMIC-IV datasets compared with SUPPORT and SYNTHETIC, implying that our model can effectively tackle high censoring rate clinical datasets, which remains a challenging task for most of the prior arts. Moreover, as shown in Figure 5, compared with DSM, which is also based on mixture model structure, our DeepSurv has a better prediction performance. It implies that dirichlet process guided mixture model can have better approach the survival curve.

| **25% Time Horizon** | SUPPORT | | SYNTHETIC | | MIMIC-III | | MIMIC-IV | |
|---|---|---|---|---|---|---|---|---|
| **Models** | C-index(%) ↑ | BS(%) ↓ | C-index(%) ↑ | BS(%) ↓ | C-index(%) ↑ | BS(%) ↓ | C-index(%) ↑ | BS(%) ↓ |
| **CPH (Cox, 1972)** | $68.52_{\pm 0.00}$ | $48.54_{\pm 0.00}$ | $62.66_{\pm 0.00}$ | $36.76_{\pm 0.00}$ | $76.41_{\pm 0.00}$ | $4.87_{\pm 0.00}$ | $71.97_{\pm 0.00}$ | $4.67_{\pm 0.00}$ |
| **DeepCox (Katzman et al., 2018)** | $69.59_{\pm 0.42}$ | $11.70_{\pm 0.05}$ | $66.98_{\pm 0.39}$ | $\underline{15.61}_{\pm 0.06}$ | $79.13_{\pm 0.91}$ | $4.00_{\pm 0.03}$ | $74.74_{\pm 0.78}$ | $4.21_{\pm 0.04}$ |
| **DeepHit (Lee et al., 2018)** | $62.90_{\pm 0.40}$ | $18.50_{\pm 1.09}$ | $61.84_{\pm 0.76}$ | $18.78_{\pm 7.77}$ | $71.49_{\pm 0.64}$ | $28.63_{\pm 1.56}$ | $70.48_{\pm 0.64}$ | $46.37_{\pm 1.60}$ |
| **DCM (Nagpal et al., 2021b)** | $\underline{76.40}_{\pm 0.99}$ | $11.58_{\pm 0.31}$ | $67.35_{\pm 0.30}$ | $15.89_{\pm 0.21}$ | $80.50_{\pm 1.16}$ | $4.05_{\pm 0.05}$ | $\underline{75.70}_{\pm 1.03}$ | $4.19_{\pm 0.04}$ |
| **DSM (Nagpal et al., 2021a)** | $75.90_{\pm 0.41}$ | $\underline{11.17}_{\pm 0.04}$ | $67.69_{\pm 0.28}$ | $15.99_{\pm 0.03}$ | $\underline{81.84}_{\pm 0.51}$ | $3.93_{\pm 0.02}$ | $75.18_{\pm 1.26}$ | $\underline{4.16}_{\pm 0.02}$ |
| **Sumo-Net (Rindt et al., 2022)** | $64.64_{\pm 2.08}$ | $28.87_{\pm 0.47}$ | $65.43_{\pm 1.75}$ | $30.90_{\pm 0.04}$ | $64.09_{\pm 0.30}$ | $22.21_{\pm 0.53}$ | $64.59_{\pm 0.13}$ | $54.85_{\pm 0.80}$ |
| **NFM (Wu et al., 2023)** | $69.91_{\pm 4.01}$ | $30.72_{\pm 0.20}$ | $\underline{67.74}_{\pm 0.44}$ | $15.34_{\pm 0.12}$ | $69.09_{\pm 0.09}$ | $59.14_{\pm 0.08}$ | $68.65_{\pm 0.02}$ | $62.57_{\pm 0.50}$ |
| **DDPSurv** | $\mathbf{76.82}_{\pm 0.34}$ | $\mathbf{11.13}_{\pm 0.03}$ | $\mathbf{68.38}_{\pm 0.38}$ | $15.85_{\pm 0.18}$ | $\mathbf{82.03}_{\pm 0.84}$ | $\mathbf{3.91}_{\pm 0.03}$ | $\mathbf{78.55}_{\pm 0.48}$ | $\mathbf{4.11}_{\pm 0.01}$ |
| **50% Time Horizon** | SUPPORT | | SYNTHETIC | | MIMIC-III | | MIMIC-IV | |
| **Models** | C-index(%) ↑ | BS(%) ↓ | C-index(%) ↑ | BS(%) ↓ | C-index(%) ↑ | BS(%) ↓ | C-index(%) ↑ | BS(%) ↓ |
| **CPH (Cox, 1972)** | $66.50_{\pm 0.00}$ | $34.34_{\pm 0.00}$ | $60.73_{\pm 0.00}$ | $23.47_{\pm 0.00}$ | $69.63_{\pm 0.00}$ | $10.23_{\pm 0.00}$ | $71.34_{\pm 00.00}$ | $11.41_{\pm 00.00}$ |
| **DeepCox (Katzman et al., 2018)** | $67.48_{\pm 0.37}$ | $19.30_{\pm 0.07}$ | $67.08_{\pm 0.40}$ | $23.13_{\pm 0.08}$ | $71.22_{\pm 1.05}$ | $9.75_{\pm 0.07}$ | $70.08_{\pm 0.54}$ | $10.47_{\pm 0.17}$ |
| **DeepHit (Lee et al., 2018)** | $63.51_{\pm 0.76}$ | $24.43_{\pm 0.47}$ | $68.09_{\pm 0.38}$ | $33.04_{\pm 9.05}$ | $71.07_{\pm 0.54}$ | $29.43_{\pm 0.74}$ | $70.28_{\pm 0.54}$ | $38.16_{\pm 0.74}$ |
| **DCM (Nagpal et al., 2021b)** | $\underline{70.76}_{\pm 0.60}$ | $19.04_{\pm 0.54}$ | $67.23_{\pm 0.11}$ | $23.62_{\pm 0.42}$ | $71.36_{\pm 0.99}$ | $9.95_{\pm 0.12}$ | $68.57_{\pm 0.47}$ | $\underline{10.58}_{\pm 0.07}$ |
| **DSM (Nagpal et al., 2021a)** | $70.19_{\pm 0.34}$ | $\underline{18.33}_{\pm 0.07}$ | $66.69_{\pm 0.28}$ | $24.10_{\pm 0.08}$ | $\underline{72.98}_{\pm 0.70}$ | $9.66_{\pm 0.06}$ | $\underline{72.86}_{\pm 1.24}$ | $10.59_{\pm 0.05}$ |
| **Sumo-Net (Rindt et al., 2022)** | $64.64_{\pm 2.08}$ | $29.94_{\pm 0.47}$ | $64.64_{\pm 2.09}$ | $32.19_{\pm 0.82}$ | $66.21_{\pm 0.35}$ | $14.26_{\pm 0.07}$ | $64.56_{\pm 0.22}$ | $36.90_{\pm 4.13}$ |
| **NFM (Wu et al., 2023)** | $63.18_{\pm 0.18}$ | $40.49_{\pm 0.15}$ | $\mathbf{69.30}_{\pm 0.21}$ | $\mathbf{20.32}_{\pm 0.12}$ | $68.67_{\pm 0.07}$ | $51.14_{\pm 0.02}$ | $67.30_{\pm 0.11}$ | $51.11_{\pm 0.03}$ |
| **DDPSurv** | $\mathbf{70.89}_{\pm 0.64}$ | $\mathbf{18.17}_{\pm 0.03}$ | $\underline{68.13}_{\pm 0.42}$ | $\underline{22.57}_{\pm 0.35}$ | $\mathbf{73.61}_{\pm 0.19}$ | $\mathbf{9.65}_{\pm 0.03}$ | $\mathbf{72.95}_{\pm 0.24}$ | $\mathbf{10.31}_{\pm 0.07}$ |

Table 2: Compared results at 25% quantile and 50% quantile time horizen. **Best results** across the comparable methods in each dataset are highlighted in bold, while the second-best results are underlined.

## 5.4 Tail performance

For evaluation of the tail performance after the heavy-tail mixture, we evaluate the performance on tail time horizon quantiles. Table 3 presents the performance of DDPSurv at the 75% and 90% quantile. It is observed that our model ranks first or second for most of the datasets and evaluation metrics, validating that our model can handle tail scenarios and mitigate the long tail bias very well. We further validate the effect of heavy-tail mixture by comparing the performance with and without mixing heavy-tail distributions, respectively. Figure 3 and Figure 4 present the results on the benchmark datasets for 0.75 time horizon and 0.9 time horizon respectively. It is observed that the model has better prediction performance with heavy-tail distribution mixed when considering C-index as evaluation metric. The observation is valid for all the datasets with Support dataset having most significant improvement.

| 75% Time Horizon | SUPPORT | | SYNTHETIC | | MIMIC-III | | MIMIC-IV | |
|---|---|---|---|---|---|---|---|---|
| **Models** | C-index(%) ↑ | BS(%) ↓ | C-index(%) ↑ | BS(%) ↓ | C-index(%) ↑ | BS(%) ↓ | C-index(%) ↑ | BS(%) ↓ |
| **CPH (Cox, 1972)** | $66.32_{\pm0.00}$ | $23.15_{\pm0.00}$ | $59.74_{\pm0.00}$ | $49.34_{\pm0.00}$ | $65.02_{\pm0.00}$ | $22.77_{\pm0.00}$ | $67.72_{\pm0.00}$ | $27.27_{\pm0.00}$ |
| **DeepCox (Katzman et al., 2018)** | $\mathbf{66.80}_{\pm0.16}$ | $\underline{22.01}_{\pm0.17}$ | $67.27_{\pm0.35}$ | $19.5_{\pm0.05}$ | $\underline{67.25}_{\pm0.83}$ | $17.99_{\pm0.20}$ | $67.64_{\pm0.81}$ | $19.72_{\pm0.12}$ |
| **DeepHit (Lee et al., 2018)** | $64.26_{\pm0.73}$ | $23.98_{\pm0.28}$ | $57.75_{\pm5.68}$ | $28.60_{\pm03.34}$ | $61.71_{\pm0.44}$ | $21.59_{\pm2.07}$ | $64.97_{\pm0.44}$ | $25.12_{\pm3.00}$ |
| **DCM (Nagpal et al., 2021b)** | $65.67_{\pm1.89}$ | $22.48_{\pm0.54}$ | $67.41_{\pm0.08}$ | $19.90_{\pm0.17}$ | $66.94_{\pm0.86}$ | $18.14_{\pm0.29}$ | $66.90_{\pm0.70}$ | $19.85_{\pm0.19}$ |
| **DSM (Nagpal et al., 2021a)** | $65.47_{\pm0.27}$ | $22.02_{\pm0.10}$ | $66.07_{\pm0.32}$ | $17.08_{\pm0.49}$ | $66.51_{\pm0.38}$ | $\underline{17.37}_{\pm0.04}$ | $67.65_{\pm1.52}$ | $\underline{19.69}_{\pm0.19}$ |
| **Sumo-Net (Rindt et al., 2022)** | $64.64_{\pm2.08}$ | $27.09_{\pm0.90}$ | $63.49_{\pm2.09}$ | $26.37_{\pm3.12}$ | $55.97_{\pm9.02}$ | $39.00_{\pm10.94}$ | $60.69_{\pm4.86}$ | $25.25_{\pm3.11}$ |
| **NFM (Wu et al., 2023)** | $63.67_{\pm0.07}$ | $34.06_{\pm0.14}$ | $\mathbf{68.54}_{\pm0.11}$ | $\mathbf{15.50}_{\pm0.06}$ | $66.28_{\pm0.23}$ | $31.90_{\pm0.03}$ | $66.48_{\pm0.26}$ | $32.36_{\pm0.02}$ |
| **DDPSurv** | $66.14_{\pm0.01}$ | $\mathbf{21.86}_{\pm0.08}$ | $\underline{67.78}_{\pm0.73}$ | $\underline{15.84}_{\pm0.29}$ | $\mathbf{67.82}_{\pm0.36}$ | $\mathbf{17.25}_{\pm0.05}$ | $\mathbf{68.42}_{\pm0.36}$ | $\mathbf{19.48}_{\pm0.10}$ |
| 90% Time Horizon | SUPPORT | | SYNTHETIC | | MIMIC-III | | MIMIC-IV | |
| **Models** | C-index(%) ↑ | BS(%) ↓ | C-index(%) ↑ | BS(%) ↓ | C-index(%) ↑ | BS(%) ↓ | C-index(%) ↑ | BS(%) ↓ |
| **CPH (Cox, 1972)** | $65.92_{\pm0.00}$ | $19.41_{\pm0.00}$ | $59.52_{\pm0.00}$ | $74.41_{\pm0.00}$ | $63.94_{\pm0.00}$ | $35.61_{\pm0.00}$ | $65.52_{\pm0.00}$ | $46.72_{\pm0.00}$ |
| **DeepCox (Katzman et al., 2018)** | $\mathbf{66.71}_{\pm0.12}$ | $17.45_{\pm0.17}$ | $6.95_{\pm0.31}$ | $10.89_{\pm0.06}$ | $65.86_{\pm0.67}$ | $22.54_{\pm0.42}$ | $65.21_{\pm1.15}$ | $23.93_{\pm0.29}$ |
| **DeepHit (Lee et al., 2018)** | $64.19_{\pm0.51}$ | $24.56_{\pm0.38}$ | $57.35_{\pm7.75}$ | $14.43_{\pm0.58}$ | $64.57_{\pm1.35}$ | $\mathbf{13.64}_{\pm0.47}$ | $64.28_{\pm0.05}$ | $\underline{14.44}_{\pm0.65}$ |
| **DCM (Nagpal et al., 2021b)** | $65.16_{\pm1.46}$ | $17.81_{\pm0.42}$ | $\underline{67.02}_{\pm0.16}$ | $10.95_{\pm0.22}$ | $65.45_{\pm0.56}$ | $22.17_{\pm0.20}$ | $64.51_{\pm1.09}$ | $23.62_{\pm0.40}$ |
| **DSM (Nagpal et al., 2021a)** | $65.10_{\pm0.20}$ | $\underline{17.36}_{\pm0.02}$ | $65.08_{\pm0.27}$ | $10.19_{\pm0.13}$ | $\underline{65.89}_{\pm0.41}$ | $21.45_{\pm0.14}$ | $64.97_{\pm1.89}$ | $23.03_{\pm0.55}$ |
| **Sumo-Net (Rindt et al., 2022)** | $64.64_{\pm2.08}$ | $28.07_{\pm0.86}$ | $63.04_{\pm3.66}$ | $57.21_{\pm3.08}$ | $55.82_{\pm8.76}$ | $42.36_{\pm9.96}$ | $60.55_{\pm4.86}$ | $\mathbf{14.18}_{\pm2.65}$ |
| **NFM (Wu et al., 2023)** | $64.34_{\pm0.08}$ | $20.07_{\pm0.05}$ | $66.85_{\pm0.11}$ | $\mathbf{9.13}_{\pm0.10}$ | $64.79_{\pm0.41}$ | $\underline{16.29}_{\pm0.00}$ | $\underline{65.56}_{\pm0.19}$ | $16.83_{\pm0.01}$ |
| **DDPSurv** | $\underline{65.95}_{\pm0.25}$ | $\mathbf{17.28}_{\pm0.05}$ | $\mathbf{67.34}_{\pm0.72}$ | $\underline{9.41}_{\pm0.13}$ | $\mathbf{66.77}_{\pm0.28}$ | $21.93_{\pm0.33}$ | $65.95_{\pm0.20}$ | $22.62_{\pm0.13}$ |

Table 3: Compared results at the tail, i.e., 75% quantile and 90% quantile. **Best results** across each dataset are in bold, while the second-best results are underlined.

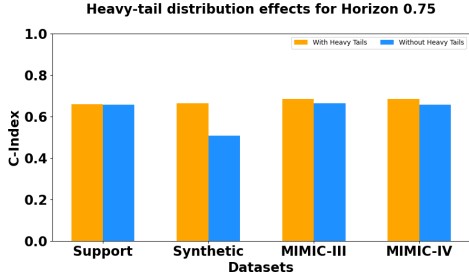

Figure 3: The performance in C-index of DDPSurv under 0.75 time horizon with and without mixing the heavy-tail distributions, respectively.

Figure 4: The performance in C-index of DDPSurv under 0.9 time horizon with and without mixing the heavy-tail distributions, respectively.

## 5.5 ABLATION ANALYSIS

**Number of mixture components.** We evaluate the effect of different numbers of mixture components $K_1$ and $K_2$ on the survival prediction performance. As shown in Figure 6, we use the C-index at 25% quantile as the evaluation metric and run our experiments on MIMIC-IV dataset with different combinations of $k_1$ and $k_2$. The results indicate that the performance generally rises as a trend when $K_1$ and $K_2$ increase within a range, which further suggests that a large number of mixture components may improve the expressiveness of our model. The results stabilize after a certain number of mixtures (e.g., $K_2 > 8$, indicating that the DP can automatically select the optimal number of mixture components, and hence reduce the reliance on tuning the number of mixtures.

**Effects of the concentration rate $\eta$.** We investigate the effect of the concentration rate on the model performance. As shown in Figure 7, we use the C-index as the evaluation metric, run our experiments on the MIMIC-III dataset and record the results for six different values of $\eta$ (we let $\eta = \eta_1 = \eta_2$) under four different test horizons. The results indicate that the concentration rate generally makes no significant impact on the performance of DDPSurv. Among these four concentration rates, the default value 10 has a slight advantage over others.

**Effects of the censoring rate.** We further investigate the effect of the censoring rate on the model performance. In previous ablation experiments, we generally use the MIMIC-III dataset to illustrate the scalability for our method. However, since the default censor rate of MIMIC-III is larger, we perform experiments on the SUPPORT dataset instead. As shown in Figure 8, we use the mean of C-index as the evaluation metric, run our experiments on the SUPPORT dataset and record the results for both our model and DeepCox, one of the outstanding baseline models. The result indicates that

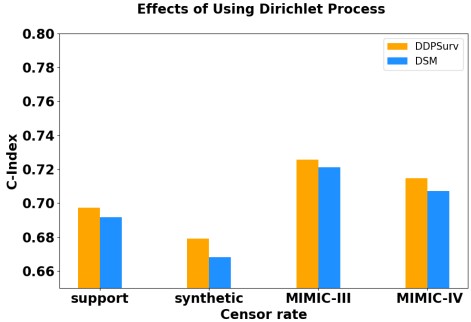

Figure 5: Comparison of the performance between DSM and our DDPSurv. We show the mean C-index of four time horizons.

Figure 6: Performance of our DDPSurv on the MIMIC-III dataset with respect to different $K_1$ and $K_2$.

our model is more robust when we adjust the censoring rate of the SUPPORT dataset, which means that our model has a stable advantage over DeepCox for most of the censoring rates.

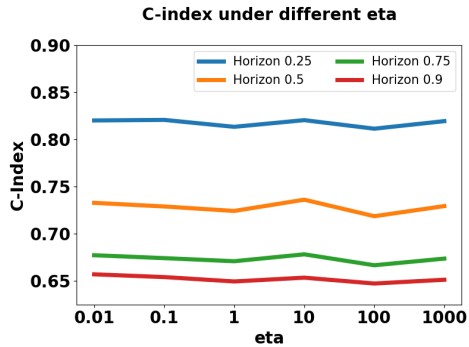

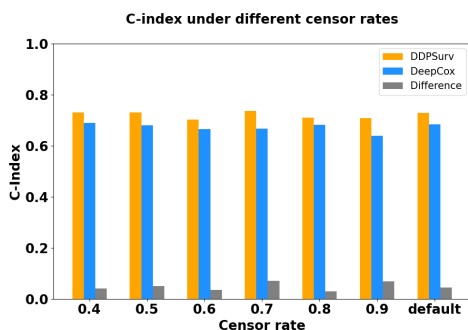

Figure 7: Performance in C-index of DDPSurv on the MIMIC-III dataset with respect to different $\eta$.

Figure 8: The mean of four time horizons' C-index under different censor rates.

## 6 CONCLUSIONS

In this work, we propose DDPSurv, a novel deep Bayesian non-parametric framework on survival prediction. By mixing heavy-tail distributions into our model, we achieve adaptive tail correction and improve the behaviour at tails. We adopt stochastic variational inference to train the model in high dimensions. Empirical results show that our method can overall outperform the baseline methods. Ablation analysis demonstrates the contribution of each proposed component and robustness to variations in hyperparameters. Our work can be potentially extended to multimodal learning, where Bayesian nonparametric methods can effectively fuse the distributions from different modalities.

**Limitations and Future Works.** One limitation of our method is that we did not explicitly model the potential heterogeneity in individuals (although implicitly by adopting a mixture model). However, DDPSurv can be easily extended to incorporate this factor with modifications in the likelihood, such as the frailty family, which will be explored in future works. Our method can also be extended to other application domains, such as survival analysis for whole slide images.

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

# A  TECHNICAL DETAILS

**Proof of Theorem 1.** We follow the FN scheme from (Wu et al., 2023) to prove the convergence properties of the proposed DDPSM model.

## A.1  TECHNICAL LEMMAS.

We list the technical lemma of the FN scheme here, where the proof of each technical lemma is extended from (Wu et al., 2023). We first develop technical lemmas for facilitating the proof of the main theorem

**Lemma 1.** Under conditions 1–3, for $(T, \delta, \boldsymbol{x}) \in [0, \tau] \times \{0, 1\} \times [-1, 1]^d$ the following terms are bounded:

1. $l(T, \delta, \boldsymbol{x}; \nu_0)$ with true parameter $\nu_0$.
2. $l(T, \delta, \boldsymbol{x}; \hat{\nu})$ with any parameter estimates $\hat{\nu}$ in any Sieve space listed in condition 2.

**Lemma 2.** Under condition 1–3, let $\hat{\nu}$, $\hat{\nu}_1$, and $\hat{\nu}_2$ be arbitrary three parameter tuples inside the sieve space defined in condition 2, then the following inequalities hold

$$\|l(T, \delta, \boldsymbol{x}; \nu_0) - l(T, \delta, \boldsymbol{x}; \hat{\nu})\|_\infty \lesssim \|\nu_0 - \hat{\nu}\|_\infty,$$
$$\|l(T, \delta, \boldsymbol{x}; \hat{\nu}_1) - l(T, \delta, \boldsymbol{x}; \hat{\nu}_1 2\|_\infty \lesssim \|\hat{\nu}_1 - \hat{\nu}_2\|_\infty.$$

**Lemma 3.** (Approximation error) For any $n$, there exists an element in the corresponding sieve space $\varpi_n \nu_0$, satisfying $d(\varpi_n \nu_0, \varpi_n \nu_0) = \mathcal{O}(n^{-\frac{\beta}{\beta+d+1}})$.

**Lemma 4.** Suppose that $\mathcal{F}$ is a class of functions satisfying that $N(\epsilon, \mathcal{F}, \|\cdot\|) < \infty$ for $\forall \epsilon > 0$. We define $\tilde{N}(\epsilon, \mathcal{F}, \|\cdot\|)$ to be the minimal number of $\epsilon$-balls $B(f, \epsilon) = \{g : \|g - f\| < \epsilon\}$ needed to cover $\mathcal{F}$ and further constrain that $f \in \mathcal{F}$. Then we have

$$N(\epsilon, \mathcal{F}, \|\cdot\|) \leq \tilde{N}(\epsilon, \mathcal{F}, \|\cdot\|) \leq N(\frac{\epsilon}{2}, \mathcal{F}, \|\cdot\|).$$

**Lemma 5.** Suppose that $\mathcal{F}$ is a class of functions satisfying $N(\epsilon, \mathcal{F}, \|\cdot\|_\infty) < \infty$ for $\forall \epsilon > 0$. We define $\tilde{N}_{[]}(\epsilon, \mathcal{F}, \|\cdot\|_\infty)$ to be the minimal number of brackets $[l, u]$ needed to cover $\mathcal{F}$ with $\|l - u\|_\infty \leq \epsilon$ and further constrain that $f \in \mathcal{F}$, $l = f - \frac{\epsilon}{2}$, and $u = f + \frac{\epsilon}{2}$. Then we have

$$N_{[]}(\epsilon, \mathcal{F}, \|\cdot\|_\infty) \leq \tilde{N}_{[]}(\epsilon, \mathcal{F}, \|\cdot\|_\infty) \leq N_{[]}(\frac{\epsilon}{2}, \mathcal{F}, \|\cdot\|_\infty).$$

**Lemma 6.** (Model Capacity) Let $\mathcal{G}_n = \{l(T, \delta, Z; \hat{v}, \hat{\delta}) : \hat{v} \in \mathcal{V}_n\}$. Under condition 2, with $s_\nu = \frac{2\beta}{2\beta+d+1}$, there exists a constant $c_\nu > 0$ such that

$$N_{[]}(\epsilon, \mathcal{G}_n), \|\cdot\|_\infty) \lesssim \frac{1}{\epsilon} N(c_\nu \epsilon^{1/s_\nu}, \mathcal{V}_n, \|\cdot\|_2).$$

We adopt the theory of empirical processes (Wellner et al., 2013; Wu et al., 2023) heavily in the proof of main theorems. The proof is based on the proof of the FN scheme introduced by (Wu et al., 2023). For a function class $\mathcal{F}$, we define $N(\epsilon, \mathcal{F}, \|\cdot\|)$ and $N_{[]}(\epsilon, \mathcal{F}, \|\cdot\|)$ to be the covering number and the bracketing number of $\mathcal{F}$ with respect to norm $\|\cdot\|$ under radius $\epsilon$, respectively. We use $\text{VC}(\mathcal{F})$ to denote the VC-dimension of $\mathcal{F}$. Moreover, we use the notation $a \lesssim b$ to denote $a \leq Cb$ for some positive constant $C$.

We define

$$l(T, \delta, \boldsymbol{x}; \hat{\nu}) = \delta \log \int_0^T e^{\nu(s, \boldsymbol{x})} ds + \delta \nu(t, \boldsymbol{x}) + \int_0^T e^{\nu(s, \boldsymbol{x})} ds.$$

Under the definition of the sieve space stated in condition 2, we restate the parameter estimate as

$$\hat{\nu}_n(t, \boldsymbol{x}) = \arg\max_{\hat{\nu} \in \mathcal{V}_n} \frac{1}{n} \sum_{i \in [n]} l(t_i, \delta_i, \boldsymbol{x}_i; \hat{\nu}).$$

Under the model assumption, $p(T, \delta | \boldsymbol{x} = \boldsymbol{x})$ can be expressed by

$$p(T, \delta | \boldsymbol{x}; \nu) = \exp(l(T, \delta, \boldsymbol{x}; v)) f_{C|\boldsymbol{x}}(T)^{1-\delta} S_{C|\boldsymbol{x}}(T)^{1-\delta}.$$

The defined distance can be explicitly expressed by

$$d(\hat{\nu}, \nu_0) = \sqrt{\mathbb{E}_{\boldsymbol{x}} \left[ \int \left| \sqrt{p(T, \delta | \boldsymbol{x}; \hat{\nu})} - \sqrt{p(T, \delta | \boldsymbol{x}; \nu_0)} \right|^2 \mu(dT \times d\delta) \right]}$$

The proof can be then divided into four steps (Wu et al., 2023)

**Step 1:**

For arbitrary $0 < \epsilon \leq 1$, we have that

$$\inf_{d(\hat{\nu}, \nu_0) \geq \epsilon} \mathbb{E}\left[ l(T, \delta, \boldsymbol{x}; \nu_0) - l(T, \delta, \boldsymbol{x}; \hat{\nu}) \right]$$

$$= \inf_{d(\hat{\nu}, \nu_0) \geq \epsilon} \mathbb{E}_{\boldsymbol{x}} \left[ \mathbb{E}_{T|\delta, \boldsymbol{x}} \left[ \log p(T, \delta | \boldsymbol{x}; \nu_0) - \log p(T, \delta | \boldsymbol{x}; \hat{\nu}_0) \right] \right]$$

$$= \inf_{d(\hat{\nu}, \nu_0) \geq \epsilon} \mathbb{E}_{\boldsymbol{x}} \left[ \text{KL}\left( \mathbb{P}_{\hat{\nu}, \boldsymbol{x}} \| \mathbb{P}_{\nu_0, \boldsymbol{x}} \right) \right]$$

Using the fact that $\text{KL}\left( \mathbb{P}_{\hat{\nu}, \boldsymbol{x}} \| \mathbb{P}_{\nu_0, \boldsymbol{x}} \right) \geq 2H^2 \left( \mathbb{P}_{\hat{\nu}, \boldsymbol{x}} \| \mathbb{P}_{\nu_0, \boldsymbol{x}} \right)$, we can further obtain that

$$\inf_{d(\hat{\nu}, \nu_0) \geq \epsilon} \mathbb{E}\left[ l(T, \delta, \boldsymbol{x}; \nu_0) - l(T, \delta, \boldsymbol{x}; \hat{\nu}) \right]$$

$$\geq \inf_{d(\hat{\nu}, \nu_0) \geq \epsilon} \mathbb{E}_{\boldsymbol{x}} \left[ 2H^2 \left( \mathbb{P}_{\hat{\nu}, \boldsymbol{x}} \| \mathbb{P}_{\nu_0, \boldsymbol{x}} \right) \right]$$

$$= 2 \inf_{d(\hat{\nu}, \nu_0) \geq \epsilon} d^2(\hat{\nu}, \nu_0)$$

$$\geq 2\epsilon^2.$$

**Step 2:** We consider the following derivations

$$\sup_{d(\hat{\nu}, \nu_0) \leq \epsilon} \text{Var}\left[ l(T, \delta, \boldsymbol{x}; \nu_0) - l(T, \delta, \boldsymbol{x}; \hat{\nu}) \right]$$

$$\leq \sup_{d(\hat{\nu}, \nu_0) \leq \epsilon} \mathbb{E}\left[ \left( l(T, \delta, \boldsymbol{x}; \nu_0) - l(T, \delta, \boldsymbol{x}; \hat{\nu}) \right)^2 \right]$$

$$= \sup_{d(\hat{\nu}, \nu_0) \leq \epsilon} \mathbb{E}_{\boldsymbol{x}} \left[ \mathbb{E}_{T|\delta, \boldsymbol{x}} \left[ \left( \log p(T, \delta, \boldsymbol{x}; \nu_0) - \log p(T, \delta, \boldsymbol{x}; \hat{\nu}_0) \right)^2 \right] \right]$$

$$= 4 \sup_{d(\hat{\nu}, \nu_0) \leq \epsilon} \mathbb{E}_{\boldsymbol{x}} \left[ \int \left( p(T, \delta, \boldsymbol{x}; \nu_0) \left( \sqrt{\frac{p(T, \delta, \boldsymbol{x}; \nu_0)}{p(T, \delta, \boldsymbol{x}; \hat{\nu}_0)}} \right)^2 \right) \mu(dT \times d\delta) \right]$$

By Taylor's expansion on $\log x$, there exists $\eta(T, \delta, \boldsymbol{x})$ between $\sqrt{p(T, \delta, \boldsymbol{x}; \nu_0)}$ and $\sqrt{p(T, \delta, \boldsymbol{x}; \hat{\nu}_0)}$ pointwisely such that

$$p(T, \delta, \boldsymbol{x}; \nu_0) \left( \log \sqrt{\frac{p(T, \delta, \boldsymbol{x}; \nu_0)}{p(T, \delta, \boldsymbol{x}; \hat{\nu}_0)}} \right)^2$$

$$= p(T, \delta, \boldsymbol{x}; \nu_0) \left( \log \sqrt{p(T, \delta, \boldsymbol{x}; \nu_0)} - \log \sqrt{p(T, \delta, \boldsymbol{x}; \hat{\nu}_0)} \right)^2$$

$$= \frac{p(T, \delta, \boldsymbol{x}; \nu_0)}{\eta(T, \delta, \boldsymbol{x})^2} \left( \sqrt{p(T, \delta, \boldsymbol{x}; \nu_0)} - \sqrt{p(T, \delta, \boldsymbol{x}; \hat{\nu}_0)} \right)^2$$

Since $p(T, \delta, \boldsymbol{x}; \nu_0)/p(T, \delta, \boldsymbol{x}; \hat{\nu}) = \exp(l(T, \delta, \boldsymbol{x}; \nu_0) - l(T, \delta, \boldsymbol{x}; \hat{\nu}))$, from lemma 1, $l(T, \delta, \boldsymbol{x}; \nu_0)$ and $l(T, \delta, \boldsymbol{x}; \hat{\nu})$ are bounded on $[0, \tau] \times \{0, 1\} \times [-1, 1]^d$ uniformly for all $\hat{\nu}$. Thus there exists constants $C_1$ and $C_2$ such that $0 < C_1 \le p(T, \delta, \boldsymbol{x}; \nu_0)/p(T, \delta, \boldsymbol{x}; \hat{\nu}) \le C_2$. This leads to the fact that $p(T, \delta, \boldsymbol{x}; \nu_0) \frac{1}{\eta(T, \delta, \boldsymbol{x})^2}$ is bounded. We further have that

$$p(T, \delta, \boldsymbol{x}; \nu_0) \left( \log \sqrt{p(T, \delta, \boldsymbol{x}; \nu_0)} - \log \sqrt{p(T, \delta, \boldsymbol{x}; \hat{\nu})} \right)^2 \lesssim \left| \sqrt{p(T, \delta, \boldsymbol{x}; \nu_0)} - \sqrt{p(T, \delta, \boldsymbol{x}; \hat{\nu})} \right|^2.$$

Thus we have that

$$\sup_{d(\hat{\nu}, \nu_0) \le \epsilon} \mathrm{Var} \left[ l(T, \delta, \boldsymbol{x}; \nu_0) - l(T, \delta, \boldsymbol{x}; \hat{\nu}) \right]$$

$$\lesssim \sup_{d(\hat{\nu}, \nu_0)} \mathbb{E}_{\boldsymbol{x}} \left[ \int \left| \sqrt{p(T, \delta, \boldsymbol{x}; \nu_0)} - \sqrt{p(T, \delta, \boldsymbol{x}; \hat{\nu})} \right|^2 \mu(dT \times d\delta) \right]$$

$$= \sup_{d(\hat{\nu}, \nu_0)} d^2(\hat{\nu}, \nu_0)$$

$$\le \epsilon^2.$$

**Step 3** We define $\tilde{\mathcal{G}}_n = \{ l(T, \delta, \boldsymbol{x}; \hat{\nu}) - l(T, \delta, \boldsymbol{x}; \gamma_n \nu_0) : \hat{\nu} \in \mathcal{V}_n \}$. Here $\varpi_n \nu_0$ has been defined in 3. Obviously, we have that $\log N_{[]}(\epsilon, \tilde{\mathcal{G}}_n, \| \cdot \|_\infty) = \log N_{[]}(\epsilon, \mathcal{G}_n, \| \cdot \|_\infty)$, where $\mathcal{G}$ is defined in lemma 6. By lemma 6, we further obtain that

$$N_{[]}(\epsilon, \mathcal{G}_n), \| \cdot \|_\infty) \lesssim \frac{1}{\epsilon} N(c_\nu \epsilon^{1/s_\nu}, \mathcal{V}_n, \| \cdot \|_2).$$

According to (Yarotsky, 2017), Theorem 7, under condition 2, we have that the VC-dimension of $\mathcal{V}_n$ satisfies that $\mathrm{VC}(\mathcal{V}_n) \lesssim n^{\frac{d+1}{\beta + d + 1}} \log^3 n \log \frac{1}{\epsilon}$. Thus we obtain that

$$\log N(c_\nu \epsilon^{1/s_\nu}, \mathcal{V}_n, \| \cdot \|) \lesssim \frac{\mathrm{VC}(\mathcal{V}_n)}{s_\nu} \log \frac{1}{\epsilon} \lesssim n^{\frac{d+1}{\beta + d + 1}} \log^3 n \log \frac{1}{\epsilon}.$$

Furthermore, we also have that $\log N_{[]}(\epsilon, \tilde{\mathcal{G}}, \| \cdot \|) \lesssim n^{\frac{d+1}{n + d + 1}} \log^3 n \log \frac{1}{\epsilon}$.

**Step 4** By the Cauchy–Schwartz inequality, we have that

$$\sqrt{\mathbb{E} \left[ l(T, \delta, \boldsymbol{x}; \hat{\nu}) - l(T, \delta, \boldsymbol{x}; \varpi_n \nu_0) \right]} \le \left[ \mathbb{E}(l(T, \delta, \boldsymbol{x}; \hat{\nu}) - l(T, \delta, \boldsymbol{x}; \varpi_n \nu_0))^2 \right]^{1/4}.$$

Then by Lemma 3 we further obtain that

$$\sqrt{\mathbb{E} \left[ l(T, \delta, \boldsymbol{x}; \hat{\nu}) - l(T, \delta, \boldsymbol{x}; \varpi_n \nu_0) \right]} \lesssim \sqrt{d(\varpi_n \nu_0, \nu_0)} \lesssim n^{-\frac{\beta}{2\beta + 2d + 2}}.$$

Now let

$$\tau = \frac{\beta}{2\beta + 2d + 2} - 2 \frac{\log \log n}{\log n}.$$

Then by steps 1, 2, 3 and Yarotsky (2017), Theorem 1,

$$d(\hat{\nu}, \nu_0) = \max\left(n^{-\tau}, d(\varpi_n\nu_0, \nu_0), \sqrt{\mathbb{E}\left[l(T, \delta, \boldsymbol{x}; \hat{\nu}) - l(T, \delta, \boldsymbol{x}; \varpi_n\nu_0)\right]}\right)$$

By lemma 3, we have $d(\varpi_n\nu_0, \nu_0) = \mathcal{O}(n^{-\frac{\beta}{\beta+d+1}})$, and by Step 4, we have $\sqrt{\mathbb{E}\left[l(T, \delta, \boldsymbol{x}; \hat{\nu}) - l(T, \delta, \boldsymbol{x}; \varpi_n\nu_0)\right]} = \mathcal{O}(n^{-\frac{\beta}{2\beta+2d+2}})$. Thus we have $d(\hat{\nu}, \nu_0) = \mathcal{O}(n^{-\frac{\beta}{2\beta+2d+2}}\log^2 n) = \tilde{\mathcal{O}}(n^{-\frac{\beta}{2\beta+2d+2}})$.

The proof of the technical lemmas is based on Wu et al. (2023).

**Proof of Lemma 1.** Since $\nu_0(T, \boldsymbol{x}) \in \mathcal{W}_M^\beta([0, \tau] \times [-1.1]^d)$, we have that $\nu_0(T, \boldsymbol{x}) \leq M$ and $\int_0^T e^{\nu(s, \boldsymbol{x})}ds \leq \tau e^M$.

$$|l(T, \delta, \boldsymbol{x}); \nu_0)|$$
$$\leq \left|\log\int_0^T e^{\nu_0(s, \boldsymbol{x})}ds\right| + |\nu_0(T, \boldsymbol{x})| + \left|\int_0^T e^{\nu_0(s, \boldsymbol{x})}ds\right|$$
$$\leq 2M + \log\tau + \tau e^M.$$

We then have that $l(T, \delta, \boldsymbol{x}; \nu_0)$ is bounded among $(T, \delta, \boldsymbol{x}) \in [0, \tau] \times \{0, 1\} \times [-1, 1]^d$. The proof of the boundedness of $l(T, \delta, \boldsymbol{x}; \hat{\nu})$ is similar.

**Proof of Lemma 2.** By definition we have that

$$|l(T, \delta, \boldsymbol{x}); \nu_0) - l(T, \delta, \boldsymbol{x}; \hat{\nu})|$$
$$\leq \left|\log\int_0^T e^{\nu_0(s, \boldsymbol{x})}ds - \log\int_0^T e^{\hat{\nu}(s, \boldsymbol{x})}ds\right| + |\nu_0(T, \boldsymbol{x}) - \hat{\nu}(T, \boldsymbol{x})|$$
$$+ \left|\int_0^T e^{\nu_0(s, \boldsymbol{x})}ds - \int_0^T e^{\hat{\nu}(s, \boldsymbol{x})}ds\right|.$$

By Taylor's expansion on $\log(\cdot)$, we can further show that

$$|l(T, \delta, \boldsymbol{x}); \nu_0) - l(T, \delta, \boldsymbol{x}; \hat{\nu})|$$
$$\leq |\nu_0(T, \boldsymbol{x}) - \hat{\nu}(T, \boldsymbol{x})| + 2\left|\int_0^T e^{\nu_0(s, \boldsymbol{x})}ds - \int_0^T e^{\hat{\nu}(s, \boldsymbol{x})}ds\right|$$

Again, by Taylor's expansion, we have

$$\left|\int_0^T e^{\nu_0(s, \boldsymbol{x})}ds - \int_0^T e^{\hat{\nu}(s, \boldsymbol{x})}ds\right| \leq \tau e^{\max(M, N_\nu)}\|\nu_0 - \hat{\nu}\|_\infty$$

Finally, we obtain that

$$|l(T, \delta, \boldsymbol{x}); \nu_0) - l(T, \delta, \boldsymbol{x}; \hat{\nu})| \leq |\nu_0(T, \boldsymbol{x}) - \hat{\nu}(T, \boldsymbol{x})| + 2\tau e^{\max(M, N_\nu)}\|\nu_0 - \hat{\nu}\|_\infty$$

Taking the supremum on both sides, we conclude that,

$$\|l(T, \delta, \boldsymbol{x}); \nu_0) - l(T, \delta, \boldsymbol{x}; \hat{\nu})\|_\infty \lesssim \|\nu_0 - \hat{\nu}\|_\infty$$

The proof of the second inequality is similar.

**Proof of Lemma 3.** According to (Yarotsky, 2017), Theorem 1, there exists an approximation function $\hat{\nu}^*$ such that $\|\nu_0 - \hat{\nu}\|_\infty = \mathcal{O}\left(n^{-\frac{\beta}{\beta+d+1}}\right)$. Let $\varpi_n\nu_0 = \hat{\nu}^*$. We have that

$$
\begin{aligned}
&d(\varpi_n\nu_0.\nu_0) \\
&= \sqrt{\mathbb{E}_{\boldsymbol{x}}\left[\int \left|\sqrt{p(T,\delta|\boldsymbol{x};\hat{\nu})} - \sqrt{p(T,\delta|\boldsymbol{x};\nu_0)}\right|^2 \mu(dT \times d\delta)\right]} \\
&= \sqrt{\mathbb{E}_{\boldsymbol{x}}\left[\int \left[e^{\frac{1}{2}l(T,\delta,\boldsymbol{x};\varpi_n\nu_0)} - e^{\frac{1}{2}l(T,\delta,\boldsymbol{x};\nu_0)}\right]^2 f_{C|\boldsymbol{x}}(T)^{1-\delta}S_{C|\boldsymbol{x}}(T)^\delta \mu(dT \times d\delta)\right]} \\
&= \left\|e^{\frac{1}{2}l(T,\delta,\boldsymbol{x};\varpi_n\nu_0)} - e^{\frac{1}{2}l(T,\delta,\boldsymbol{x};\nu_0)}\right\|_\infty \sqrt{\mathbb{E}_{\boldsymbol{x}}\left[\int f_{C|\boldsymbol{x}}(T)^{1-\delta}S_{C|\boldsymbol{x}}(T)^\delta\right]}
\end{aligned}
$$

By lemmas 1 and 2, we have that

$$
\begin{aligned}
\left\|e^{\frac{1}{2}l(T,\delta,\boldsymbol{x};\varpi_n\nu_0)} - e^{\frac{1}{2}l(T,\delta,\boldsymbol{x};\nu_0)}\right\|_\infty &\le \|\varpi_n\nu_0 - \nu_0\|_\infty \\
&= \mathcal{O}\left(n^{-\frac{\beta}{\beta+d+1}}\right).
\end{aligned}
$$

Since $f_{C|\boldsymbol{x}}(T)^{1-\delta} \le f_{C|\boldsymbol{x}}(T)$ and $S_{C|\boldsymbol{x}}(T)^\delta \le 1$, we also have that

$$
\begin{aligned}
\sqrt{\mathbb{E}_{\boldsymbol{x}}\left[\int f_{C|\boldsymbol{x}}(T)^{1-\delta}S_{C|\boldsymbol{x}}(T)^\delta \mu(dT \times d\delta)\right]} &\le \sqrt{\mathbb{E}\left[(1 + f_{C|\boldsymbol{x}}(T))\mu(dT \times d\delta)\right]} \\
&\le \sqrt{2 + 2\tau}.
\end{aligned}
$$

Thus we obtain that $d(\varpi_n\nu_0.\nu_0) = \mathcal{O}\left(n^{-\frac{\beta}{\beta+d+1}}\right)$.

**Proof of Lemma 4 and Lemma 5.** Omitted as the proof is similar to (Wu et al., 2023).

**Proof of Lemma 6.** By lemma 5, first we have that $N_{[]}(\epsilon, \mathcal{G}_n, \|\cdot\|_\infty) \le \tilde{N}_{[]}(\epsilon, \mathcal{G}_n, \|\cdot\|_\infty)$. By lemma 2, there exists a constant $c_1 > 0$ such that for arbitrary $\hat{\nu}_1, \hat{\nu}_2 \in \mathcal{V}_n$, we have that

$$
\|l(T,\delta,\boldsymbol{x});\hat{\nu}_1) - l(T,\delta,\boldsymbol{x};\hat{\nu}_2)\|_\infty \le c_1\|\hat{\nu}_1 - \hat{\nu}_2\|_\infty,
$$

which indicates that as long as $\|\hat{\nu}_1 - \hat{\nu}_2\|_\infty \le \frac{\epsilon}{2c_3}$, we have that $\|l(T,\delta,\boldsymbol{x});\hat{\nu}_1) - l(T,\delta,\boldsymbol{x};\hat{\nu}_2)\|_\infty \le \epsilon$. Thus, we have

$$
\tilde{N}_{[]}(\epsilon, \mathcal{G}_n, \|\cdot\|_\infty) \le \tilde{N}_{[]}(\frac{\epsilon}{2c_3}, \mathcal{V}_n, \|\cdot\|_\infty).
$$

## B   VARIATIONAL UPDATES OF MIXTURE WEIGHTS.

We present more details on updating the variational mixture weights. We have the following closed-form solution of $\gamma$ that minimizes the KL divergence term

$$
\gamma_{1,k} = 1 + \sum_{b=1}^{B}\phi_{b,k}, \quad \gamma_{2,k} = \eta_1 + \sum_{b=1}^{B}\sum_{r=k+1}^{T}\phi_{b,r}, \tag{10}
$$

for $b \in \{1, \ldots, B\}$, where $B$ is the sample size and $T$ is the maximum number of clusters. We then compute the log of posterior responsibility (i.e., the weighted $\log\phi$) as follows,

## C   ADDITIONAL DETAILS ON DATASETS.

## D   DETAILS OF DISTRIBUTIONS

We specify the distributions used in this work and their useful properties.

### D.1 PRIMITIVE DISTRIBUTIONS

We provide the density and survival functions of the primitive distributions.

**Weibull Distribution.**

The density of the Weibull distribution is given by:

$$f(x; \lambda, k) = \frac{k}{\lambda} \left(\frac{x}{\lambda}\right)^{k-1} e^{-(x/\lambda)^k},$$

The survival function of Weibull distribution is given by:

$$F(x; \lambda, k) = 1 - e^{-(x/\lambda)^k}$$

where $k > 0$ is the shape parameter and $\lambda > 0$ is the scale parameter.

**Log-normal Distribution.**

The density of the Log-normal distribution is given by:

$$f(x|\mu, \sigma) = \frac{1}{x\sigma\sqrt{2\pi}} e^{-\frac{(\ln x - \mu)^2}{2\sigma^2}}$$

The survival function of Log-normal distribution is given by:

$$F(x|\mu, \sigma) = \frac{1}{2} + \frac{1}{2}\text{erf}\left(\frac{\ln x - \mu}{\sqrt{2}\sigma}\right)$$

where $\mu$ is the shape parameter and $\sigma > 0$ is the scale parameter.

The error function erf(x) is defined as:

$$\text{erf}(x) = \frac{2}{\sqrt{\pi}} \int_0^x e^{-t^2} dt$$

### D.2 HEAVY-TAIL DISTRIBUTIONS

**Log-Cauchy Distribution.** We adopt the log-Cauchy distribution as the heavy tail distribution.

The multivariate Gaussian Distribution is defined as

$$p(\boldsymbol{x}; \boldsymbol{\mu}, \boldsymbol{\Sigma}) = \frac{1}{(2\pi)^{\frac{n}{2}} |\boldsymbol{\Sigma}|^{\frac{1}{2}}} \exp\left\{-\frac{1}{2}(\boldsymbol{x} - \boldsymbol{\mu})^T \boldsymbol{\Sigma}^{-1}(\boldsymbol{x} - \boldsymbol{\mu})\right\}$$

### D.3 KL DIVERGENCES OF TWO MULTIVARIATE NORMAL DISTRIBUTION

The KL divergences of two multivariate normal distributions $\mathcal{N}(\boldsymbol{\mu}_1, \Sigma_1)$ and $\mathcal{N}(\boldsymbol{\mu}_2, \Sigma_2)$

$$\text{KL}(\mathcal{N}(\boldsymbol{\mu}_1, \boldsymbol{\Sigma}_1) \| \mathcal{N}(\boldsymbol{\mu}_2, \boldsymbol{\Sigma}_2)) = \frac{1}{2}\left[\log\frac{|\boldsymbol{\Sigma}_2|}{|\boldsymbol{\Sigma}_1|} - p + \text{tr}\{\boldsymbol{\Sigma}_2^{-1}\boldsymbol{\Sigma}_1\} + (\boldsymbol{\mu}_2 - \boldsymbol{\mu}_1)^T \boldsymbol{\Sigma}_2^{-1}(\boldsymbol{\mu}_2 - \boldsymbol{\mu}_1)\right]$$

## E MORE ON BASELINE METHODS AND IMPLEMENTATION DETAILS

### E.1 IMPLEMENTATION DETAILS AND HYPERPARAMETERS

We present additional implementation details and hyperparameter settings. We first provide the key settings and adaptations applied to the baseline methods for reproducibility. We follow the default settings for other fine-grained parameters (e.g., learning rates).

The proposed method is implemented in Python with *Pytorch* library on a server equipped with four NVIDIA GeForce RTX 3090 GPUs.

All models are pre-trained with 10000 iterations and then trained with 100 epochs with possible early stopping. We use the *Adam* optimizer to optimize the model with a learning rate of $1 \times 10^{-4}$.

### E.2 DETAILED DESCRIPTIONS ON EVALUATION METRICS

- The concordance index or the C-index is a generalization of the area under the ROC curve (AUC) that can take into account censored data. It represents the global assessment of the model discrimination power: this is the model's ability to correctly provide a reliable ranking of the survival times based on the individual risk scores.

- The Brier Score is a strictly proper score function or strictly proper scoring rule that measures the accuracy of probabilistic predictions. For uni-dimensional predictions, it is strictly equivalent to the mean squared error as applied to predicted probabilities.

