# OpenReview forum: "Bayesian Nonparametric Survival Analysis via Deep Dirichlet Process"
_ICLR.cc/2025/Conference — ICLR 2025 Conference Withdrawn Submission_

### Official Review · Reviewer_RMUD · 2024-11-02

**Soundness:** 3
**Presentation:** 3
**Contribution:** 2
**Rating:** 5
**Confidence:** 4

**Summary:**

This paper proposes the type of DP mixtures for survival analysis. The paper follows the standard approach of the well-known DPM (Dirichlet Process Mixture). DPM is not typical since the proposed DPM has two kernels (mixtures of two types of distributions such as log-normal and log-Cauchy, denoted as primitive distribution). The stick-breaking process and ELBO are used for the Bayesian inference. Theoretical properties of consistency with respect to $W_M^{\beta}$ were validated in a standard form.

Experiments were conducted to show the superiority and properties of the proposed algorithm. Considered datasets are MIMIC III and V, and various algorithms with DNN and metrics such as C-index, Brier score, and others are examined. The experiments include datasets of SUPPORT, Synthetic, and MIMIC-III/IV. It shows the superiority of the proposed algorithm and analytic results for the tail probability. Various ablation studies are done for the validation of hyper-parameters' effect.

**Strengths:**

The use of DP with two kernels is an interesting strategy. It has the potential to address the heavy tail probability. The proposed approach is nonparametric survival analysis, which can address the arbitrary distributions in survival analysis. Also, the theoretical validation makes this paper more solid.

**Weaknesses:**

$\bullet$
At first, experiments are somewhat limited in datasets. Besides the considered datasets, there are various datasets such as SEER, GBSG, METABRIC, etc. More datasets are required to validate the generalization.

$\bullet$
I don’t know the effect of the usage of a heavy-tailed kernel. This paper lacks of delicate analysis of the design of DP. To validate the effect of a heavy-tailed kernel, experiments with one kernel or controlling the ratio of kernels by $k_1$ and $k_2$ are required. It seems that the performance gain is not impressive, especially in the 90% quantile time horizon.

**Questions:**

$\bullet$
Usually, the C-index can be calculated for overall survival times. It can be better to clarify the C-index calculation on the time horizon in a math formulation.

$\bullet$ In theoretical aspects, are the results not affected by the kernels (log-normal or log-Cauchy)?

$\bullet$ In my opinion, the ratio of $k_1$ and $k_2$, $r$ can be another hyper-parameter. What do you think about this setup?

---

### Official Review · Reviewer_KERF · 2024-11-03

**Soundness:** 2
**Presentation:** 1
**Contribution:** 2
**Rating:** 3
**Confidence:** 4

**Summary:**

The authors propose a mixture of "primitive" distributions (Weibull, log-normal), as well as long-tailed distributions with mixing weights obtained by a Dirichlet Process. The covariates are first transformed by applying a deep neural network and predict the parameters of the distributions in the mixture, making this method neural network based. The authors use stochastic VI on a mean-field ELBO to optimize the parameters and find the posterior distribution. The main advantages of this approach is adding a long-tail to the distribution, which is deemed to be important in recent survival literature, as well as reducing the number of hyper-parameters, as the number of mixtures is chosen by the DP. The authors add a proof of convergence of a model where the hazard rate is predicted by a neural network and the likelihood is optimized. The empirical performance against many SOTA methods is shown and an ablation study is added to show the relevance of the chosen architecture.

**Strengths:**

- The model seems to perform very well, however I have not checked reproducibility
- Ablatation studies are valuable to show the influence of the heavy tails
- Reducing the number of hyperparameters in the formulation is valuable

**Weaknesses:**

- The paper is not easy to understand, notations could be more clear, some statements are contradictory and there are many typos and errors, which should be fixed. Parts of the appendix are missing.
- I don't understand how the proof is relevant to the model in the paper. The authors assume the hazard rate is given by a neural network and use the survival loss in the proof, but the actual model is a mixture of distributions with an ELBO optimization. I might be missing something here, but it seems just based on established literature and irrelevant.
- If the proof is relevant, the implications of the proof could be explained better and it should be explained how this is distinguishing the model from other models.
- Dirichlet Process Mixture models for survival is not novel, nor is adding long tailed distributions (cure-models). Encoding the parameters with a NN and using VI, while seemingly working well, is in my opinion not novel enough for this venue.

**Questions:**

- Deephit does not use variational inference for their loss, but the full loss likelihood (lines 90/91)
- The authors write variational EM in figure 1, but variational inference is used
- Should the conditional of the last distribution of equation 5 and 7 not depend on alpha/beta and the conditional of the first term $\xi$ or $\zeta$, depending on if k is in heavy-tail or short tail? The notation is slightly confusing here.
- Appendix B is missing
- In the ablation study for censoring, why does a stable advantage (the difference between c-index is the same over censoring rate) imply that the model is *more* robust to censoring? Would that not mean that they are as robust with the DeepSurv model just underperforming in comparison?

---

### Official Review · Reviewer_LVX9 · 2024-11-04

**Soundness:** 2
**Presentation:** 1
**Contribution:** 2
**Rating:** 3
**Confidence:** 4

**Summary:**

The authors propose a framework for time-to-event modeling called DDPSurv which models the survival distribution as a Dirichlet process mixture. They show that by mixing heavy-tailer distributions, their approach can better handle long-tail bias and that their formulation readily extends to competing risks. Theoreitical results are presented regarding the asymptotic bound to the posterior concentration rate of the model parameters. Experiments on MIMIC illustrate the capabilities of the proposed approach.

**Strengths:**

The authors consider the problem of modeling time-to-event distribution using a nonparametric mixture based on the DP process mixture.

The ablation studies are welcome.

**Weaknesses:**

The methodological components of the paper are very difficult to follow. Specifically:
- Figures 1 and 2, and Algorithm 1 are not mentioned or discussed in the text.
- Some quantities have not been defined, e.g., \tilde{X}, p(Z=k|x,beta), Z, \phi. Some can be guessed from the figures though.
- Equation (5) needs further explanation. Where is zeta, zeta are the parameters of (2) but (2) is not in (5), there are only K_1 \zeta values but they are indexed to K_1+K_2?
- Algorithm 1 describes a collection of networks that are not described.

The Theoretical results in Section 4 are welcome but are completely decontextualized of the rest of the paper.

The datasets need to be described in more detail, specially, the synthetic dataset and MIMIC. The former for being introduced in the paper and the latter from being a general dataset and survival tasks can be specified in a multitude of ways.

In the main experiment (Table 2) it is not clear why C-indices are not presented over the complete time horizon or at least done over time. Further, some of the gains by the proposed model are not statistically significant after accounting for variation.

Results in Table 3 are underwhelming because one will expect the proposed model which is specifically designed for long tails to more clearly outperform other models on long horizons, however, that does not seem the case. In fact the proposed model seems to perform better on lower quantiles (comparing Table 2 and 3).

Though it is good to see ablation studies for the number of mixture components, concentration rate and censoring rate, these are only selectively compared to a few other modeling approaches and largely disconnected from the tail behavior and performance of the model.

**Questions:**

See weaknesses.

---

### Official Review · Reviewer_U7jd · 2024-11-05

**Soundness:** 3
**Presentation:** 3
**Contribution:** 1
**Rating:** 5
**Confidence:** 3

**Summary:**

This paper considered using deep bayes non-parameter analysis in via the
deep dirichlet process. A key feature different from previous work is the new capacity to process the long tail survival approximation and very high-dimensional datasets. Extensive experimental results verify the effectiveness of the proposed approach.

**Strengths:**

I would think this paper addresses the interesting points in modern survival analysis. With experiments in real healthcare datasets, this paper properly justified the practical implications of this proposed approach.

**Weaknesses:**

In general, I feel the major weakness comes from the significance section, where I find it’s very hard to differentiate the key differences between the current method and related work.

1. I feel like this looks like a simple adaptation of well-known technology. Therefore I could not identify the actual novelty or significance of current submission. For example, using the dirichlet process, or non-parameter bayes are not new in the related work (e.g, [1-5]). This reviewer finds it very difficult to know the actual new contributions, compared with previous large amounts of related papers. For example, in [3], it is a book to systematically discuss dirichlet process and survival analysis. I would think authors should make more clear explanations on the key novelties.

2. The two key contributions, handling high-dimensional data and long tail survival approximation are seemingly not effectively justified. For example, the high-dimensional data is incorporated with a regular deep learning model, where it seems no very high-dimensional data (such as medical images) are tested. Besides, the mean-field approximation makes it a bit hard to believe how this can indeed tackle the long-tail.


References

1. Neural Survival Clustering: Non-parametric mixture of neural networks for survival clustering. 2022
2. Bayesian nonparametric Erlang mixture modeling for survival analysis. 2023
3. Dirichlet Process – with Applications on Survival Analysis. Zhaoheng Li and Yeheng Zong. 2021-12-15
4. Reliable survival analysis based on the Dirichlet process. 2015
5. Semi-parametric survival analysis via Dirichlet process mixtures of the First Hitting Time model. 2021

**Questions:**

I would like authors to address the significance and novelty concerns.

---

### Note · Authors · 2024-11-19

**Comment:**

We appreciate the reviewers for their valuable comments. We will further enhance this manuscript based on your suggestions.

**Withdrawal Confirmation:**

I have read and agree with the venue's withdrawal policy on behalf of myself and my co-authors.